# High-Frequency Signal Injection-Based Sensorless Control for Dual-Armature Flux-Switching Permanent Magnet Machine

**Lijian Wu, Jiali Yi, Zekai Lyu** **, Zhengxiang Zhang and Sideng Hu \***

College of Electrical Engineering, Zhejiang University, Hangzhou 310027, China; ljw@zju.edu.cn (L.W.)
\* Correspondence: husideng@zju.edu.cn

**Abstract:** The new topology of the dual-armature flux-switching permanent magnet machine (DA-FSPM) leads to new characteristics and issues in the control of the machine, of which the mutual inductance of the two sets of armature windings is the most important one. This paper proposes a novel position–sensorless control method based on high-frequency injection (HFI) for DA-FSPM. The high-frequency model of the machine is derived, and the theory of the position estimation method is proposed. Different from the conventional HFI-based position estimation method, the proposed method utilizes the mutual inductance of the DA-FSPM rather than the machine saliency. Meanwhile, because the extracted position information based on the mutual inductance is more obvious, the proposed method also has better steady and dynamic performance. Then, the position observer based on the phase lock loop and the initial position detection method for the DA-FSPM is proposed. The experiments are executed on a DA-FSPM prototype with three-phase stator windings and five-phase rotor windings to prove the effectiveness and superiority of the proposed method.

**Keywords:** dual-armature flux-switching permanent magnet machine (DA-FSPM); high-frequency injection (HFI); mutual inductance; position estimation; sensorless control



## 1. Introduction

Due to the high power density and efficiency, the FSPM machine is becoming a greater point of concern in industrial applications, such as aerospace and electric vehicle industries [1,2]. The DA-FSPM shown in Figure 1 is proposed in [3,4], which furthermore improved the torque density and fault-tolerant capability of the FSPM machine by adding a set of armature windings on the rotor teeth. With the improved power density and fault-tolerant capability, the application prospect of the DA-FSPM is widened.

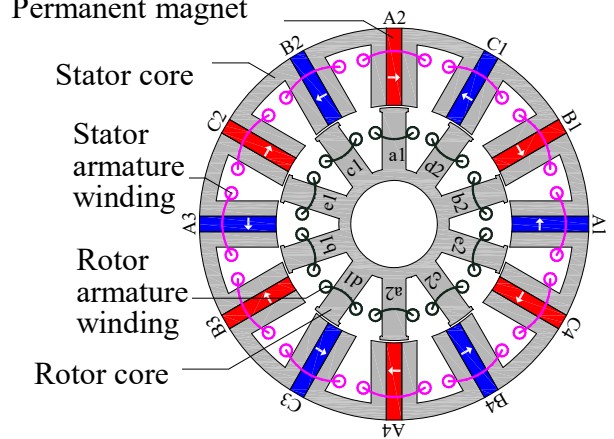

**Figure 1.** Topology of the DA-FSPM.

Although the machine design theory was researched and proposed in [1,2], the control method of the DA-FSPM is rarely discussed. The decoupled vector control for the DA-FSPM is proposed in [5], which considered and solved the cross-coupling of the stator windings and rotor windings by matrix transformation. In [5], an encoder was used to acquire the position of the machine, which increases the complexity and cost of the machine drive system [6,7]. Thus, to simplify the implementation of the drive system, the sensorless vector control for the DA-FSPM is researched in this paper.

The traditional sensorless vector control methods can be roughly divided into two categories, the first is the model-based methods, and the second is the saliency-based methods [8]. The former methods can be further divided into open-loop methods and closed-loop methods. The open loop method is based on the mathematical model of the machine, which has a high dynamic response speed [9], but it is also highly dependent on the parameters [10]. Thus, for higher robustness and accuracy, closed-loop back-EMF estimation methods were preferred [11,12]. Moreover, the position observers are based on PLL [13], Luenberger observer [14], linear state observers [15], SMO [16], EKF [17], and so on, and they were developed for higher estimation accuracy and better dynamic performance. However, the above model-based methods are normally applied in the medium- and high-speed range, while the saliency-based methods are adopted more for the zero- and low-speed region. HFI methods are widely utilized to track saliency [10]. The basic thought of the HFI method is to inject a high-frequency voltage (current) signal into the machine to excite a high-frequency current (voltage) signal, which can be used to extract the rotor position by appropriate procession [18]. The HFI can be subdivided into the rotating signal injection [19] and pulsating signal injection [20]. The former is easier to implement, and the latter has several advantages, such as less dependent on saliency, better performance in coping with dead time, better dynamic performance, and less torque ripple and additional loss [21]. Two major issues regarding HFI methods have recently been widely researched: high-frequency noise and filter abuse. As for the noise issue, several research works have been carried out over the decades. In [22], the frequency of the injected voltage was increased to the PWM frequency that is near the audible range to eliminate the noise. The noise reduction method based on pseudo-random HFI was proposed in [23,24]. Multi methods were proposed to deal with problems such as a decrease in bandwidth and estimation accuracy. The dual quasi-resonate controller was used in [25] to substitute the traditional Butterworth filter, which reduced the estimation error. Additionally, [26] proposed a novel heterodyne method to extract the position error, which removed the filters in the HFI method, while [27] utilized the neural network adaptive filter to avoid the usage of the bandpass filter in the signal demodulation. As mentioned above, a single sensorless method cannot cope with the full speed range, which leads to the research of the hybrid method. The HFI was combined with the voltage model-based position estimator in [28], while in [29], the EMF was used to combine with the HFI. Moreover, the transition methods from low speed to high speed were also important for full-speed sensorless control. The weighting factors were used for the transition from low-speed methods to high-speed methods; in [30], the speed weighting factor was used, and the position weighting factor was adopted in [31].

As discussed, the conventional HFI-based sensorless vector control methods have poor performance on surface-mounted or low-salience machines. As shown in [3], the differential inductance of the DA-FSPM is not obvious; thus, the conventional methods need modification for better performance. The HFI based on magnetic saliency for the surface-mounted machines by injecting fluctuating high-frequency voltage signals on the *d*-axis was proposed in [32]. However, the magnetic saliency is not obvious; thus, the position estimation was more difficult. Another HFI-based sensorless vector control for the dual-three PMSM, which used the zero-sequence voltage, was proposed in [33,34]. The method utilized the resistor networks and neutral points of the machine to acquire the zero-sequence voltage and then extract the position information. However, the introduction of the resistor networks increases the cost and decreases the efficiency of the drive system.

In conclusion, the existing HFI-based sensorless control method is either based on saliency or introduced additional components.

Thanks to the mutual inductance of the DA-FSPM, which is much larger than the differential inductance, the HFI-based sensorless control is modified for the machine in this paper. The conventional HFI sensorless vector control methods are extended to the DA-FSPM, and a position estimation method based on the mutual inductance of the stator windings and rotor windings rather than the machine saliency is proposed, which is the main contribution of the work. Additionally, the high-frequency model of the DA-FSPM and the high-frequency current responses under the high-frequency pulsating voltage injection are derived considering the mutual inductance of the stator windings and rotor windings. Then, the feasibility of estimating position through the current response of the same side windings and opposite side windings is discussed, and their performances are compared. Moreover, the position observer and the initial position detection method for the DA-FSPM are proposed. In conclusion, a novel HFI-based sensorless vector control method based on the mutual inductance of the DA-FSPM is proposed, which has better steady and dynamic performance compared to the traditional saliency-dependent methods.

This paper is organized as follows. In Section 2, the fundamental mathematical model of the DA-FSPM is derived, and the high-frequency model considering the mutual inductance of the machine is further established. Based on the high-frequency model, the current responses with the high-frequency pulsating voltage injection are calculated in Section 3. Then, the feasibility of extracting position information from both windings is discussed in Section 4, which also proposes the position observer and the initial position detection method for both stator windings and rotor windings positions. The experiments are carried out on a DA-FSPM prototype with three-phase stator windings and five-phase rotor windings to verify the superiority of the proposed method.

## 2. Modelling of the Machine

### 2.1. The Fundamental Frequency Model of the Machine

The DA-FSPM prototype studied in this paper is with star-connected three-phase stator windings and five-phase rotor windings. Neglecting the non-ideal factors, such as the saturation effect, eddy current loss, and hysteresis loss, the mathematical model of the DA-FSPM in the stationary coordinate system is constructed. The vector control is normally used for obtaining decoupled currents in the machine control field, which is also used in this paper. For the DA-FSPM, the three-phase Park transformation matrix and the five-phase vector space decomposition transformation matrix are respectively applied on the three-phase stator windings and five-phase rotor windings. Thus, the fundamental-frequency mathematical model of which is derived as (1) and (2) [4].

$$\begin{bmatrix} \mathbf{u_{s\_dq}} \\ \mathbf{u_{r\_d1q1d3q3}} \end{bmatrix} = \begin{bmatrix} \mathbf{R_s} & \mathbf{0} \\ \mathbf{0} & \mathbf{R_r} \end{bmatrix} \begin{bmatrix} \mathbf{i_{s\_dq}} \\ \mathbf{i_{r\_d1q1d3q3}} \end{bmatrix} + p \begin{bmatrix} \boldsymbol{\psi_{s\_dq}} \\ \boldsymbol{\psi_{r\_d1q1d3q3}} \end{bmatrix} + \begin{bmatrix} \mathbf{W_s} & \mathbf{0} \\ \mathbf{0} & \mathbf{W_r} \end{bmatrix} \begin{bmatrix} \boldsymbol{\psi_{s\_dq}} \\ \boldsymbol{\psi_{r\_d1q1d3q3}} \end{bmatrix} \quad (1)$$

$$\begin{bmatrix} \boldsymbol{\psi_{s\_dq}} \\ \boldsymbol{\psi_{r\_d1q1d3q3}} \end{bmatrix} = \begin{bmatrix} \mathbf{L_{s\_dq}} & (\mathbf{5/2})\mathbf{M_{srdq}} \\ (\mathbf{3/2})\mathbf{M_{srdq}^T} & \mathbf{L_{r\_d1q1d3q3}} \end{bmatrix} \begin{bmatrix} \mathbf{i_{s\_dq}} \\ \mathbf{i_{r\_d1q1d3q3}} \end{bmatrix} + \begin{bmatrix} \boldsymbol{\psi_{fs\_dq}} \\ \boldsymbol{\psi_{fr\_d1q1d3q3}} \end{bmatrix} \quad (2)$$

The inductance and the cross-coupling coefficient matrices are shown as follows.

$$\begin{bmatrix} \mathbf{L_{sdq}} & (\mathbf{5/2})\mathbf{M_{srdq}} \\ (\mathbf{3/2})\mathbf{M_{srdq}^T} & \mathbf{L_{rdq}} \end{bmatrix} = \begin{bmatrix} L_{sd} & 0 & \frac{5}{2}M_{srd1} & 0 & \frac{5}{2}M_{srd3} & 0 \\ 0 & L_{sq} & 0 & \frac{5}{2}M_{srq1} & 0 & \frac{5}{2}M_{srq3} \\ \frac{3}{2}M_{srd1} & 0 & L_{rd1} & 0 & M_{rd13} & 0 \\ 0 & \frac{3}{2}M_{srq1} & 0 & L_{rq1} & 0 & M_{rq13} \\ \frac{3}{2}M_{srd3} & 0 & M_{rd13} & 0 & L_{rd3} & 0 \\ 0 & \frac{3}{2}M_{srq3} & 0 & M_{rq13} & 0 & L_{rq3} \end{bmatrix} \quad (3)$$

$$
\begin{bmatrix} \mathbf{W_s} & \mathbf{0} \\ \mathbf{0} & \mathbf{W_r} \end{bmatrix} = \begin{bmatrix} 0 & -\omega_{se} & 0 & 0 & 0 & 0 \\ \omega_{se} & 0 & 0 & 0 & 0 & 0 \\ 0 & 0 & 0 & -\omega_{re} & 0 & 0 \\ 0 & 0 & \omega_{re} & 0 & 0 & 0 \\ 0 & 0 & 0 & 0 & 0 & -3\omega_{re} \\ 0 & 0 & 0 & 0 & 3\omega_{re} & 0 \end{bmatrix} \tag{4}
$$

However, the coupling still exists between the stator windings and rotor windings as (3) still contains the non-diagonal element. Thus, for the vector control of the DA-FSPM, the further decoupling method proposed in [5] is adopted in this paper. The diagonalized voltage equation (5) can be obtained through the transformation matrix in (6).

$$
\begin{bmatrix} \mathbf{u_{s\_dq}} \\ \mathbf{u_{r\_d1q1d3q3}} \end{bmatrix} = diag\left(L_{sd}, L_{sq}, L_{rd1}, L_{rq1}, L_{rd3}, L_{rq3}\right) p \begin{bmatrix} \mathbf{I_{s\_dq}} \\ \mathbf{I_{r\_d1q1d3q3}} \end{bmatrix} + \begin{bmatrix} \mathbf{R_s} & \mathbf{0} \\ \mathbf{0} & \mathbf{R_r} \end{bmatrix} \begin{bmatrix} \mathbf{I_{s\_dq}} \\ \mathbf{I_{r\_d1q1d3q3}} \end{bmatrix} + \begin{bmatrix} \mathbf{u_{s\_feedforword}} \\ \mathbf{u_{r\_feedforword}} \end{bmatrix} \tag{5}
$$

where $\mathbf{I_{s\_dq}}$ is the decoupled currents as in (7); the $\mathbf{u_{s\_feedforward}}$ and $\mathbf{u_{r\_feedforward}}$ are shown in (8).

$$
\mathbf{T} = \begin{bmatrix} 1 & 0 & \frac{5M_{srd1}}{2L_{sd}} & 0 & \frac{5M_{srd3}}{2L_{sd}} & 0 \\ 0 & 1 & 0 & \frac{5M_{srq1}}{2L_{sq}} & 0 & \frac{5M_{srq3}}{2L_{sq}} \\ \frac{3M_{srd1}}{2L_{rd1}} & 0 & 1 & 0 & \frac{M_{rd13}}{L_{rd1}} & 0 \\ 0 & \frac{3M_{srq1}}{2L_{rq1}} & 0 & 1 & 0 & \frac{M_{rq13}}{L_{rq1}} \\ \frac{3M_{srd3}}{2L_{rd3}} & 0 & \frac{M_{rd13}}{L_{rd3}} & 0 & 1 & 0 \\ 0 & \frac{3M_{srq3}}{2L_{rq3}} & 0 & \frac{M_{rq13}}{L_{rq3}} & 0 & 1 \end{bmatrix} \tag{6}
$$

$$
\begin{bmatrix} \mathbf{I_{s\_dq}} \\ \mathbf{I_{r\_d1q1d3q3}} \end{bmatrix} = \mathbf{T} \begin{bmatrix} \mathbf{i_{s\_dq}} \\ \mathbf{i_{r\_d1q1d3q3}} \end{bmatrix} \tag{7}
$$

$$
\begin{aligned}
\begin{bmatrix} \mathbf{u_{s\_feedforword}} \\ \mathbf{u_{r\_feedforword}} \end{bmatrix} &= \begin{bmatrix} \mathbf{W_s} & \mathbf{0} \\ \mathbf{0} & \mathbf{W_r} \end{bmatrix} \begin{bmatrix} \mathbf{L_{sdq}} & \frac{5}{2}\mathbf{M_{srdq}} \\ \frac{3}{2}\mathbf{M_{srdq}^T} & \mathbf{L_{rdq}} \end{bmatrix} \begin{bmatrix} \mathbf{i_{s\_dq}} \\ \mathbf{i_{r\_d1q1d3q3}} \end{bmatrix} \\
&+ \begin{bmatrix} \mathbf{W_s} & \mathbf{0} \\ \mathbf{0} & \mathbf{W_r} \end{bmatrix} \begin{bmatrix} \mathbf{\psi_{fs\_dq}} \\ \mathbf{\psi_{fr\_d1q1d3q3}} \end{bmatrix} + \begin{bmatrix} \mathbf{R_s} & \mathbf{0} \\ \mathbf{0} & \mathbf{R_r} \end{bmatrix} [\mathbf{I_6} - \mathbf{T}] \begin{bmatrix} \mathbf{i_{s\_dq}} \\ \mathbf{i_{r\_d1q1d3q3}} \end{bmatrix}
\end{aligned} \tag{8}
$$

From (5), the coefficient matrix of the current differential term is a diagonal matrix; thus, the currents can be controlled individually. Based on the decoupled model in (5) to (8), the vector control scheme for the DA-FSPM can be further constructed.

### 2.2. The High-Frequency Model of the Machine

For further analysis of HFI, the high-frequency model of the DA-FSPM is constructed in this section. Based on the fundamental frequency model, the high-frequency model of the DA-FSPM can be represented as in (9).

$$
\begin{cases}
\begin{bmatrix} \mathbf{u_{sh\_dq}} \\ \mathbf{u_{rh\_d1q1d3q3}} \end{bmatrix} = \begin{bmatrix} \mathbf{R_s} & \mathbf{0} \\ \mathbf{0} & \mathbf{R_r} \end{bmatrix} \begin{bmatrix} \mathbf{i_{sh\_dq}} \\ \mathbf{i_{rh\_d1q1d3q3}} \end{bmatrix} + p \begin{bmatrix} \mathbf{\psi_{s\_dq}} \\ \mathbf{\psi_{r\_d1q1d3q3}} \end{bmatrix} + \begin{bmatrix} \mathbf{W_s} & \mathbf{0} \\ \mathbf{0} & \mathbf{W_r} \end{bmatrix} \begin{bmatrix} \mathbf{\psi_{s\_dq}} \\ \mathbf{\psi_{r\_d1q1d3q3}} \end{bmatrix} \\
\begin{bmatrix} \mathbf{\psi_{s\_dq}} \\ \mathbf{\psi_{r\_d1q1d3q3}} \end{bmatrix} = \begin{bmatrix} \mathbf{L_{sh\_dq}} & (5/2)\mathbf{M_{srhdq}} \\ (3/2)\mathbf{M_{srhdq}^T} & \mathbf{L_{rh\_d1q1d3q3}} \end{bmatrix} \begin{bmatrix} \mathbf{i_{sh\_dq}} \\ \mathbf{i_{rh\_d1q1d3q3}} \end{bmatrix} + \begin{bmatrix} \mathbf{\psi_{fs\_dq}} \\ \mathbf{\psi_{fr\_d1q1d3q3}} \end{bmatrix}
\end{cases} \tag{9}
$$

where $\mathbf{u_{sh\_dq}}$ and $\mathbf{u_{rh\_d1q1d3q3}}$ are the injected high-frequency voltages in the rotational coordinate system; $\mathbf{i_{sh\_dq}}$ and $\mathbf{i_{rh\_d1q1d3q3}}$ are the high-frequency response currents in the rotational coordinate system; $\mathbf{L_{sh\_dq}}$, $\mathbf{L_{rh\_d1q1d3q3}}$, and $\mathbf{M_{srhdq}}$ are the high-frequency inductance matrices which are same as the fundamental frequency inductances, respectively.

The high-frequency voltage injection method is normally applied in working conditions of low speed or zero speed. Thus, to simplify the high-frequency model, the back-EMF item and the voltage drop on the resistance in the voltage equation are omitted,

which are far less than the voltage drops on the high-frequency resistance. The simplified high-frequency voltage equation is shown in (10).

$$
\begin{bmatrix} \mathbf{u_{sh\_dq}} \\ \mathbf{u_{rh\_d1q1d3q3}} \end{bmatrix} = \begin{bmatrix} \mathbf{L_{sh\_dq}} & (5/2)\mathbf{M_{srhdq}} \\ (3/2)\mathbf{M_{srhdq}^{T}} & \mathbf{L_{rh\_d1q1d3q3}} \end{bmatrix} p \begin{bmatrix} \mathbf{i_{sh\_dq}} \\ \mathbf{i_{rh\_d1q1d3q3}} \end{bmatrix}
$$ (10)

As demonstrated in (10), due to the mutual inductance, if the high-frequency voltage is injected in the stator windings or the windings, not only does the same side generate the high-frequency currents but also the opposite side. Thus, the position estimation can be through both the same side and the opposite side, which is analyzed in the next section.

## 3. High-Frequency Injection

Compared to the conventional FSPM machine, the DA-FSPM has two sets of armature windings, which means that the high-frequency voltage can be injected through the stator windings or the rotor windings. In addition, as shown in (10), the machine has differential inductance and unneglectable mutual inductance simultaneously. Thus, both injection conditions are discussed in this section, as well as the high-frequency current responses. Moreover, in this paper, the pulsating high-frequency voltage injection is adopted due to the advantages discussed before.

### 3.1. The High-Frequency Current Response

The closed-loop position estimating is based on extracting the position error information from the currents; thus, the high-frequency current responses relating to the position error are derived in this section. The estimated stator windings' electrical angle is denoted as $\hat{\theta}_{se}$ and the real stator windings electrical angle is $\theta_{se}$, and the same is true for the rotor windings.

Therefore, the transformation matrix from the estimated rotational coordinate system to the real rotational coordinate system can be obtained as follows

$$
\begin{cases}
\begin{bmatrix} f_{sdh} \\ f_{sqh} \end{bmatrix} = \mathbf{T_{s\delta}} \cdot \begin{bmatrix} f'_{sdh} \\ f'_{sqh} \end{bmatrix} = \begin{bmatrix} \cos(\delta_{se}) & \sin(\delta_{se}) \\ -\sin(\delta_{se}) & \cos(\delta_{se}) \end{bmatrix} \begin{bmatrix} f'_{sdh} \\ f'_{sqh} \end{bmatrix} \\
\begin{bmatrix} f_{rdh1} \\ f_{rqh1} \\ f_{rdh3} \\ f_{rqh3} \end{bmatrix} = \mathbf{T_{r\delta}} \cdot \begin{bmatrix} f'_{rdh1} \\ f'_{rqh1} \\ f'_{rdh3} \\ f'_{rqh3} \end{bmatrix} = \begin{bmatrix} \cos(\delta_{re}) & \sin(\delta_{re}) & 0 & 0 \\ -\sin(\delta_{re}) & \cos(\delta_{re}) & 0 & 0 \\ 0 & 0 & \cos(3\delta_{re}) & \sin(3\delta_{re}) \\ 0 & 0 & -\sin(3\delta_{re}) & \cos(3\delta_{re}) \end{bmatrix} \begin{bmatrix} f'_{rdh1} \\ f'_{rqh1} \\ f'_{rdh3} \\ f'_{rqh3} \end{bmatrix}
\end{cases}
$$ (11)

As aforementioned, the high-frequency currents in the real rotational coordinate system can be obtained by (10), which is combined with (11) to calculate the high-frequency currents in the estimated rotational coordinate system can be derived as

$$
\begin{bmatrix} i'_{sdh} \\ i'_{sqh} \\ i'_{rdh1} \\ i'_{rqh1} \\ i'_{rdh3} \\ i'_{rqh3} \end{bmatrix} = \begin{bmatrix} \mathbf{T_{s\delta}^{-1}} & 0 \\ 0 & \mathbf{T_{r\delta}^{-1}} \end{bmatrix} \begin{bmatrix} i_{sdh} \\ i_{sqh} \\ i_{rdh1} \\ i_{rqh1} \\ i_{rdh3} \\ i_{rqh3} \end{bmatrix} = \begin{bmatrix} \mathbf{T_{s\delta}^{-1}} & 0 \\ 0 & \mathbf{T_{r\delta}^{-1}} \end{bmatrix} \cdot \mathbf{L_h^{-1}} \cdot \begin{bmatrix} \mathbf{T_{s\delta}} & 0 \\ 0 & \mathbf{T_{r\delta}} \end{bmatrix} \cdot \int \begin{bmatrix} u'_{sdh} \\ u'_{sqh} \\ u'_{rdh1} \\ u'_{rqh1} \\ 0 \\ 0 \end{bmatrix} = \mathbf{A} \int \begin{bmatrix} u'_{sdh} \\ u'_{sqh} \\ u'_{rdh1} \\ u'_{rqh1} \\ 0 \\ 0 \end{bmatrix}
$$ (12)

where **A** is the coefficient matrix that relates to the position error and inductances of the DA-FSPM, which is given in detail in Appendix A. The injection in the third harmonic plane of the five-phase rotor windings is not considered in this paper.

As matrix **A** contains $\delta_{se}$ and $\delta_{re}$, some conclusions can be drawn from (12). When a high-frequency voltage signal is injected into the stator or the rotor side, the position angle signal will be carried in the current response of the two-sided winding. The current response on the same side of the injected high-frequency voltage side only carries the position information of that side, while the current response on the opposite side carries

the position information of both sides. However, as the initial position of both windings are identified, the position of the stator windings and rotor winding has a certain relationship. Thus, the position can be estimated through both sides regardless of which side the high-frequency voltage is injected.

### 3.2. Injected through the Stator Side

The high-frequency pulsating voltage is injected in the $\hat{d}$-axis of the stator windings in this paper. The high-frequency currents omitting the third harmonic component of the five-phase rotor windings can be calculated from (12) as:

$$
\begin{bmatrix} i'_{sdh} \\ i'_{sqh} \\ i'_{rdh1} \\ i'_{rqh1} \end{bmatrix} = \frac{V_c}{\omega_h} \sin\theta_h \begin{bmatrix} \hat{L_{sd}}\cos^2(\delta_{se}) + \hat{L_{sq}}\sin^2(\delta_{se}) \\ \frac{\hat{L_{sd}} - \hat{L_{sq}}}{2}\sin(2\delta_{se}) \\ \frac{3}{2}\left(\hat{M_{srd1}} + \hat{M_{srq1}}\right)\cos(\delta_{se} - \delta_{re}) + \frac{3}{2}\left(\hat{M_{srd1}} - \hat{M_{srq1}}\right)\cos(\delta_{se} + \delta_{re}) \\ \frac{3}{2}\left(\hat{M_{srd1}} + \hat{M_{srq1}}\right)\sin(\delta_{re} - \delta_{se}) + \frac{3}{2}\left(\hat{M_{srd1}} - \hat{M_{srq1}}\right)\sin(\delta_{se} + \delta_{re}) \end{bmatrix} \quad (13)
$$

where $\hat{L_{sd}}$, $\hat{L_{sq}}$, $\hat{M_{srd1}}$, and $\hat{M_{srq1}}$ are the parameters related to the inductances of the DA-FSPM, which are given in Appendix A.

As shown in (13), the amplitude of the response currents in the stator windings and rotor windings both contain information on the position angle error, which agrees with the analysis before. However, there are differences between them. The amplitude of the current response of the stator windings varies at the frequency of twice the stator electrical angle frequency; meanwhile, the varied frequency of the rotor windings' current response amplitude is related to the number of the pole and slot of the stator and rotor. The frequency of the stator winding's current response is related to the differential inductance, while it of the rotor windings is related to the mutual inductance. Thus, it is difficult to estimate the position from the same side of the saliency of the machine is not obvious, the same as the surface-mounted PMSM. However, the estimation from the opposite side is not affected by the saliency.

### 3.3. Injected through the Rotor Side

Similarly, the high-frequency voltage is injected in the $\hat{d}_1$-axis of the rotor side, and the current response can be calculated as in (14).

$$
\begin{bmatrix} i'_{sdh} \\ i'_{sqh} \\ i'_{rdh1} \\ i'_{rqh1} \end{bmatrix} = \frac{V_c}{\omega_h} \sin\theta_h \begin{bmatrix} \frac{5}{2}\left(\hat{M_{srd1}} + \hat{M_{srq1}}\right)\cos(\delta_{se} - \delta_{re}) + \frac{5}{2}\left(\hat{M_{srd1}} - \hat{M_{srq1}}\right)\cos(\delta_{se} + \delta_{re}) \\ \frac{5}{2}\left(\hat{M_{srd1}} + \hat{M_{srq1}}\right)\sin(\delta_{se} - \delta_{re}) + \frac{5}{2}\left(\hat{M_{srd1}} - \hat{M_{srq1}}\right)\sin(\delta_{se} + \delta_{re}) \\ \hat{L_{rd1}}\cos^2(\delta_{re}) + \hat{L_{rq1}}\sin^2(\delta_{re}) \\ \frac{\hat{L_{rd1}} - \hat{L_{rq1}}}{2}\sin(2\delta_{re}) \end{bmatrix} \quad (14)
$$

Similar conclusions can be drawn from (14); when the high-frequency pulsating voltage is injected in the rotor windings, the high-frequency currents related to the position information generate on both sides. Moreover, the amplitude of the high-frequency currents is related to the phase numbers of the windings, as shown in (13) and (14).

## 4. Position Estimation

As analyzed before, the current responses of both sides contain the position information no matter which side the high-frequency voltage is injected into. Thus, the position can be estimated through the same side or the opposite side, which is analyzed in this section.

### 4.1. Estimate from the Same Side

As shown in the current responses, (13) and (14), the *d-q* axes currents corresponding to the high-frequency voltage injection side both contain the position error information. However, the *d*-axis current also contains a dc component; thus, it is more convenient

to extract the position information from the $q$-axis current, which can be represented as follows:

$$i'_{iqh} = \frac{V_c}{2\omega_h} k \sin(2\delta_{ie}) \sin\theta_h \tag{15}$$

where

$$k = \begin{cases} \hat{L_{sd}} - \hat{L_{sq}}, & i = s \\ \hat{L_{rd1}} - \hat{L_{rq1}}, & i = r \end{cases} \tag{16}$$

where $i = s$ denotes that the high-frequency voltage is injected through the stator windings and $i = r$ denotes that the high-frequency voltage is injected through the rotor windings.

To extract the position information through (15), the high-frequency reference signal is used to demodulate the current response, and a low-pass filter is used to obtain the dc component which contains the position error, which can be represented as

$$\begin{aligned} f(\delta_{ie}) &= LPF\left( i'_{iqh} \sin\theta_h \right) \\ &= LPF\left( \frac{V_c}{4\omega_h} k \sin(2\delta_{ie})(1 - \sin(2\theta_h)) \right) \\ &= \frac{V_c}{4\omega_h} k \sin(2\delta_{ie}) \end{aligned} \tag{17}$$

When the position error is small enough, (17) can be rewritten as

$$f(\delta_{ie}) \approx \frac{V_c}{2\omega_h} k \delta_{ie} \tag{18}$$

Therefore, the approximate linear function of the position error can be obtained by demodulation and filtering, and the position can be estimated by the position observer.

*4.2. Estimate from the Opposite Side*

As shown in (13) and (14), the $d$-axis current responses of the opposite side are cosine, while the q-axis current responses are sine. Thus, the $q$-axis current responses in (19) are used to extract the position information for the convenience of approximation.

$$\begin{bmatrix} i'_{rqh1} \\ i'_{sqh} \end{bmatrix} = \frac{V_c}{\omega_h} \sin\theta_h \begin{bmatrix} \frac{3}{2}\left(\hat{M_{srd1}} + \hat{M_{srq1}}\right)\sin(\delta_{re} - \delta_{se}) + \frac{3}{2}\left(\hat{M_{srd1}} - \hat{M_{srq1}}\right)\sin(\delta_{se} + \delta_{re}) \\ \frac{5}{2}\left(\hat{M_{srd1}} + \hat{M_{srq1}}\right)\sin(\delta_{se} - \delta_{re}) + \frac{5}{2}\left(\hat{M_{srd1}} - \hat{M_{srq1}}\right)\sin(\delta_{se} + \delta_{re}) \end{bmatrix} \tag{19}$$

where $i'_{rqh1}$ is the high-frequency current response of the rotor windings when the high-frequency voltage is injected through the stator windings, and $i'_{sqh}$ is the high-frequency current response of the stator windings when the high-frequency voltage is injected through the rotor windings.

Then, the high-frequency reference signal and low-pass filter are used to demodulate and obtain the dc component with the position error, as follows:

$$\begin{aligned} f(\delta_{ie}) &= LPF\left( \begin{bmatrix} i'_{rqh1} \\ i'_{sqh} \end{bmatrix} \cdot \sin\theta_h \right) \\ &= \frac{V_c}{4\omega_h} \begin{bmatrix} 3\left(\hat{M_{srd1}} + \hat{M_{srq1}}\right)\sin(\delta_{re} - \delta_{se}) + 3\left(\hat{M_{srd1}} - \hat{M_{srq1}}\right)\sin(\delta_{se} + \delta_{re}) \\ 5\left(\hat{M_{srd1}} + \hat{M_{srq1}}\right)\sin(\delta_{se} - \delta_{re}) + 5\left(\hat{M_{srd1}} - \hat{M_{srq1}}\right)\sin(\delta_{se} + \delta_{re}) \end{bmatrix} \end{aligned} \tag{20}$$

The (20) can be furthermore written as (24) when the position error is small enough.

$$\begin{aligned} f(\delta_{ie}) &= \frac{V_c}{4\omega_h} \begin{bmatrix} 3\left(\hat{M_{srd1}} + \hat{M_{srq1}}\right)(\delta_{re} - \delta_{se}) + 3\left(\hat{M_{srd1}} - \hat{M_{srq1}}\right)(\delta_{se} + \delta_{re}) \\ 5\left(\hat{M_{srd1}} + \hat{M_{srq1}}\right)(\delta_{se} - \delta_{re}) + 5\left(\hat{M_{srd1}} - \hat{M_{srq1}}\right)(\delta_{se} + \delta_{re}) \end{bmatrix} \\ &= \frac{V_c}{2\omega_h} \begin{bmatrix} 3\hat{M_{srd1}}\delta_{re} - 3\hat{M_{srq1}}\delta_{se} \\ 5\hat{M_{srd1}}\delta_{se} - 5\hat{M_{srq1}}\delta_{re} \end{bmatrix} \end{aligned} \tag{21}$$

As discussed before, the actual position can be obtained through the position observer.

Of note, both the stator and rotor positions are needed for the vector control of the DAFSPM. When the initial position of the stator windings and rotor windings are detected, the actual positions of the two sets of windings can be obtained by transferring (18) and (21) into the estimate of the mechanical position of the DA-FSPM.

### 4.3. The Position Observer

#### 4.3.1. The Position Error Function

As aforementioned, the position error function $f(\delta)$ that has a linear relationship with the electrical angle error can be extracted through the high-frequency current response on the same and opposite sides after the demodulation and filtering. The estimated position can be obtained by the $f(\delta)$ and the position observer, which is discussed in this section.

As for the position error function extracted through the same side, considering that the initial position is detected, (18) can be rewritten as

$$f(\delta_m) \approx \frac{V_c}{2\omega_h}kp_i\delta_m = k_{err0}\delta_m \tag{22}$$

Meanwhile, the position error function extracted through the opposite side can also be represented as

$$f(\delta_m) = -3\left(\hat{M_{srd1}}p_r + \hat{M_{srq1}}p_s\right)\frac{V_c}{2\omega_h}\delta_m = k_{err1}\delta_m \tag{23}$$

$$f(\delta_m) = 5\left(\hat{M_{srd1}}p_s + \hat{M_{srq1}}p_r\right)\frac{V_c}{2\omega_h}\delta_m = k_{err2}\delta_m \tag{24}$$

(23) is the position error function under the stator windings injection and (24) is under the rotor windings injection. As shown in (22) to (24), whether it is a stator windings injection or a rotor windings injection and whether the position error function is extracted from the same side or the response from a opposite side, when the position error is small, the position error function can be transformed into a linear function related to the position error. Therefore, the position observer can be designed.

#### 4.3.2. The Phase-Locked Loop

In this paper, the PLL with PI controller, as shown in Figure 2, is used, where $k_{err}$ represents the slope of the position error function.

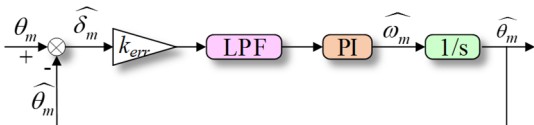

**Figure 2.** Position observer.

Thus, the transfer function of the position observer can be derived as

$$\frac{\hat{\theta_m}}{\theta_m} = \frac{k_{err}\omega_c\left(k_p s + k_i\right)}{s^3 + \omega_c s^2 + k_{err}\omega_c k_p s + k_{err}\omega_c k_i} \tag{25}$$

where $\omega_c$ is the cut-off frequency of the low-pass filter, $k_p$ is the proportional coefficient, and $k_i$ is the integral coefficient of the PI controller.

To keep the closed-loop system stable, $k_p$ and $k_i$ are calculated as

$$\begin{cases} k_p = \frac{\omega_c}{3k_{err}} \\ k_i = \frac{\omega_c^2}{27k_{err}} \end{cases} \tag{26}$$

Therefore, the transfer function of the system can be rewritten as

$$\frac{\hat{\theta}_m}{\theta_m} = \frac{\frac{1}{3}\omega_c^2 s + \frac{1}{27}\omega_c^3}{s^3 + \omega_c s^2 + \frac{1}{3}\omega_c^2 s + \frac{1}{27}\omega_c^3} \tag{27}$$

### 4.3.3. The Effect of Current Sample Error on Position Observation

The extract of the position error function is based on the current response; thus, the current sample error can affect the position observation. The error caused by the current sample can be treated as a disturbance, then the closed-loop system can be represented as in Figure 3.

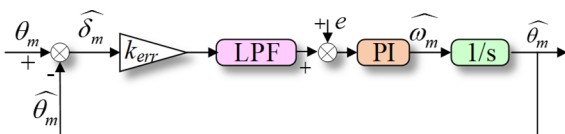

**Figure 3.** The closed-loop system, considering the current sampling error.

Combining the PI parameters in Section 4.3.2, the transfer function of estimated speed, estimated position, and current sampling disturbance input can be derived as

$$\frac{\hat{\omega}_m}{e} = \frac{9\omega_c s + \omega_c^2}{27 k_{err} s} \tag{28}$$

$$\frac{\hat{\theta}_m}{e} = \frac{9\omega_c s + \omega_c^2}{27 k_{err} s^2} \tag{29}$$

The relationship between the error of the speed estimation and the current sampling disturbance is proportional integral. The proportional coefficient and integral coefficient are consistent with the parameter design in the PI-type LLP, and they are all inversely proportional to the position error function coefficient $k_{err}$. Based on the above analysis, the relationship between the error of the speed estimation and the position estimation, and the current sampling disturbance is affected by the error function coefficient $k_{err}$. The smaller the $k_{err}$ is, the error generated in the estimated value is larger. Thus, when the current sampling accuracy is the same, and the PI controller parameters of the position observer are the same, the position estimation method with a smaller $k_{err}$ should be adopted to improve the accuracy of the estimation.

### 4.4. The Initial Position Detection

As aforementioned, the detection of the initial position is important in the sensorless control, which can affect the startup of the machine. For the DA-FSPM, the initial position of the stator windings and the rotor windings both need to be detected. Similar to position estimation, the detection of the initial position can also be achieved by injecting high-frequency voltage into one side. In this paper, the high-frequency voltage is injected into the stator windings, and the initial positions of the stator windings and rotor windings are detected through the corresponding current responses. The detection can also be achieved by injecting high-frequency voltage into rotor windings, which is not repeated in this paper.

#### 4.4.1. The Initial Position of the Stator Windings

The initial position of the stator windings can be detected by (18), and the designed position observer as the machine is static. When (18) equals zero, the initial position of the stator windings is obtained. However, there exist four possible solutions when (18) equals zero, which should be corrected for detection. When the estimated angle $\hat{\theta}_{se} = 0$, the error function of the stator windings position can be drawn as Figure 4.

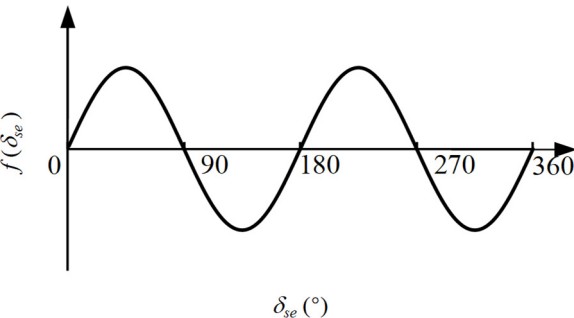

**Figure 4.** Error function of the stator windings position.

Thus, the detected initial position may have an error of $\pi$ when the actual position is at the interval of $(\pi/2, 3\pi/2)$, which means the polarity judgment is wrong. In order to compensate for the error, the saturation characteristics of the magnetic field are used in this paper to make the polarity judgment.

As shown in Figure 5, considering the saturation effect, when the positive voltage is applied on the stator $d$-axis, the $d$-axis inductance will decrease as the current increases. When the negative voltage is applied on the stator $d$-axis, the $d$-axis inductance will increase as the current decreases. Therefore, according to the saturation characteristics, a polarity judgment can be made.

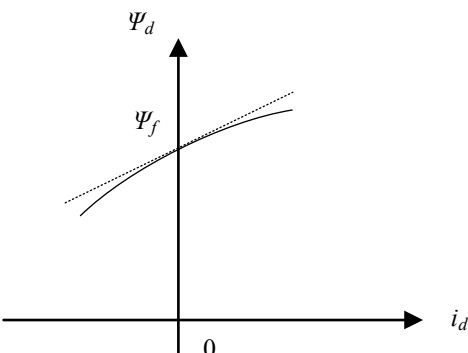

**Figure 5.** The relationship between $\Psi_d$ and $i_d$.

4.4.2. The Initial Position of the Rotor Windings

When the initial position of the stator windings is detected, (21) can be rewritten as

$$f(\delta_{re}) = 3\hat{M}_{srd1} \frac{V_c}{2\omega_h} \sin(\delta_{re})$$

(30)

Similarly, the error function of the rotor position can be drawn as Figure 6.

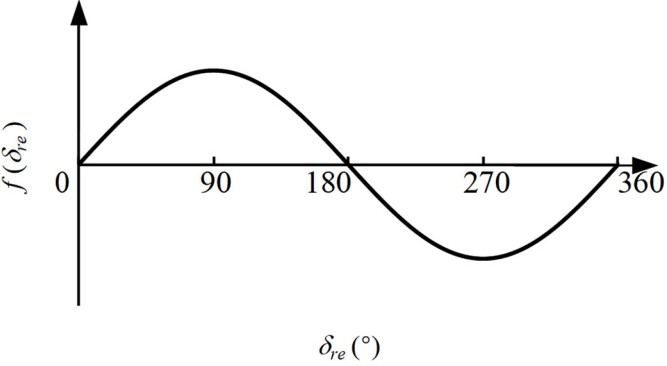

**Figure 6.** Error function of the rotor windings position.

As shown in Figure 6, no matter which interval is in the real position, the estimated position error can converge to 0 by the adjustment of the PI controller, which means that there is no need for the polarity judgment for the detection of the initial position of the rotor windings.

### 4.4.3. The Control Scheme of the Initial Position Detection

Above all, the control scheme of the initial position detection can be concluded in Figure 7. The high-frequency voltage is injected into the stator windings, and then the initial positions of the stator windings and rotor windings are detected through the corresponding currents and position observers. The initial position of the stator windings obtained through the position observer is then compensated by the polarity judgment, and the initial position of the rotor windings does not need compensation. The control scheme of the initial position detection based on the rotor side high-frequency voltage injection has the same principle as the stator side injection; thus, it is not repeated here.

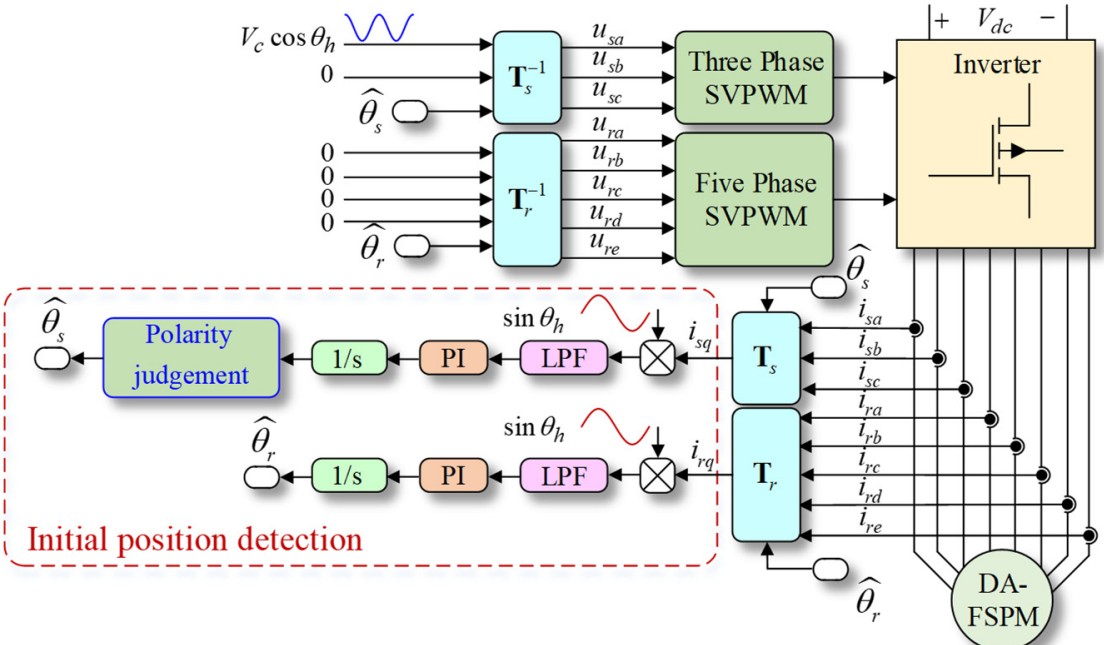

**Figure 7.** The initial position detection scheme based on the stator side high-frequency voltage injection.

### 4.5. The Overall Control Scheme

The overall control scheme of the sensorless vector control for DA-FSPM based on HFI can be concluded in Figure 8, which shows the situation of injecting to the stator windings. As shown in the scheme, the stator and rotor positions can be estimated through the current response of the same side or the opposite side with the same position observer designed before. The speed loop with PI controller is omitted in the scheme, and the $id = 0$ strategy is adopted in this paper. The control scheme of signal injection into the rotor windings is similar to Figure 8 and is not repeated here.

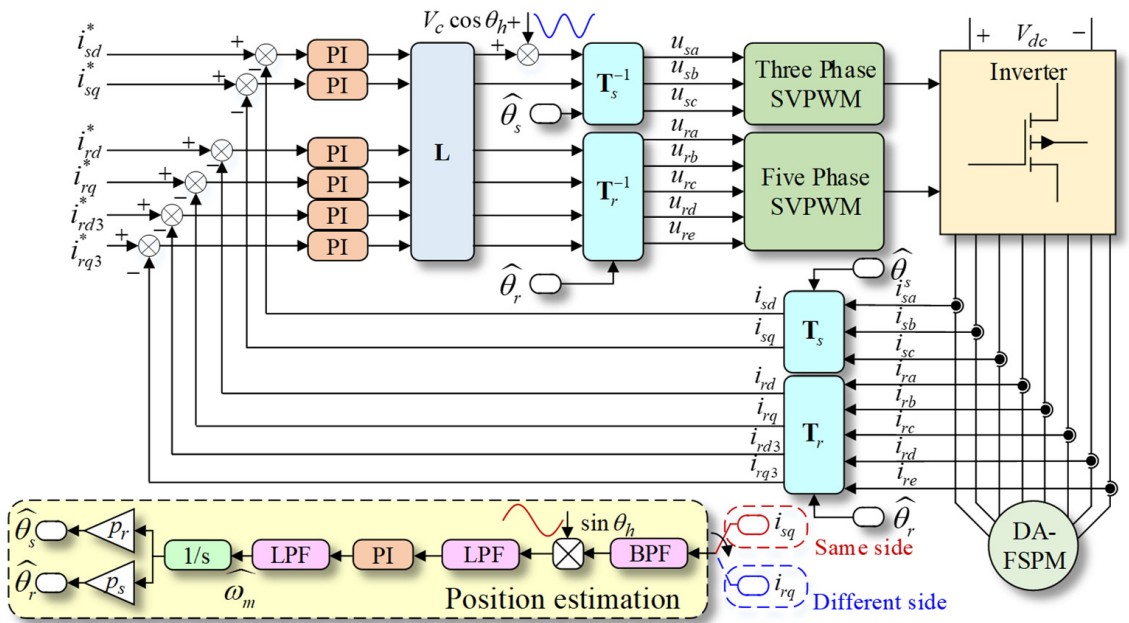

**Figure 8.** The sensorless vector control for DA-FSPM based on high-frequency voltage injection.

## 5. Experiment Results

### 5.1. The Experiment Platform

The experiment is carried out on a DA-FAPM prototype with the parameters in Table 1, and the experiment platform is shown in Figure 9. The dynamometer that connects to the DA-FSPM provides the load torque, and the 4096PR encoder provides the actual position for comparing the accuracy of the position estimation. The algorithm is implemented on the control suit, which is composed of a DC voltage source, a multi-phase inverter (8-phases are used in the experiment), and a controller with Texas Instruments TMS320F28346 DSP. In this paper, the voltage with a frequency of 1 kHz and amplitude of 10 V is injected into the stator windings.

**Table 1.** Parameters of the DA-FSPM prototype.

| Parameter | Value | Parameter | Value |
|---|---|---|---|
| $n_{ps}$ | 10 | $n_{pr}$ | 6 |
| $R_s$ (Ω) | 0.7 | $R_r$ (Ω) | 0.66 |
| $L_{sd}$ (mH) | 4.9605 | $L_{rd1}$ (mH) | 6.3961 |
| $L_{sq}$ (mH) | 5.4473 | $L_{rq1}$ (mH) | 7.1630 |
| $M_{srd1}$ (mH) | 1.7208 | $L_{rd3}$ (mH) | 3.6681 |
| $M_{srq1}$ (mH) | 1.6906 | $L_{rq3}$ (mH) | 3.6974 |
| $M_{srd3}$ (mH) | 0.4332 | $M_{srq3}$ (mH) | 0.4768 |
| $\psi_{fsd}$ (Wb) | 0.058 | $\psi_{frd1}$ (Wb) | 0.0923 |
| | | $\psi_{frd3}$ (Wb) | $-0.0065$ |

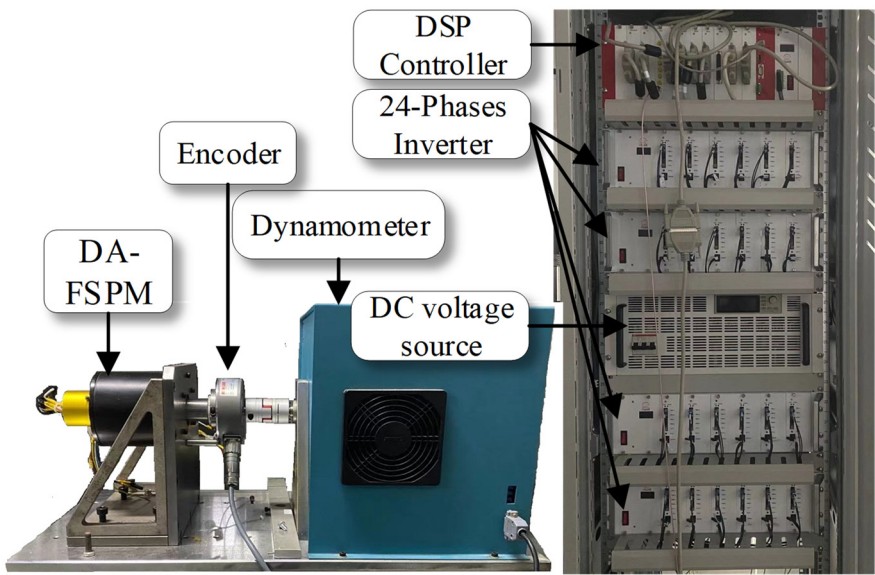

**Figure 9.** The test platform of the DA-FSPM prototype.

*5.2. Initial Position Detection Results*

The experiment results of the initial position detection with the initial position set as 0.4 rad, 0.8 rad, 1.2 rad, and 1.5 rad are shown in Figure 10.

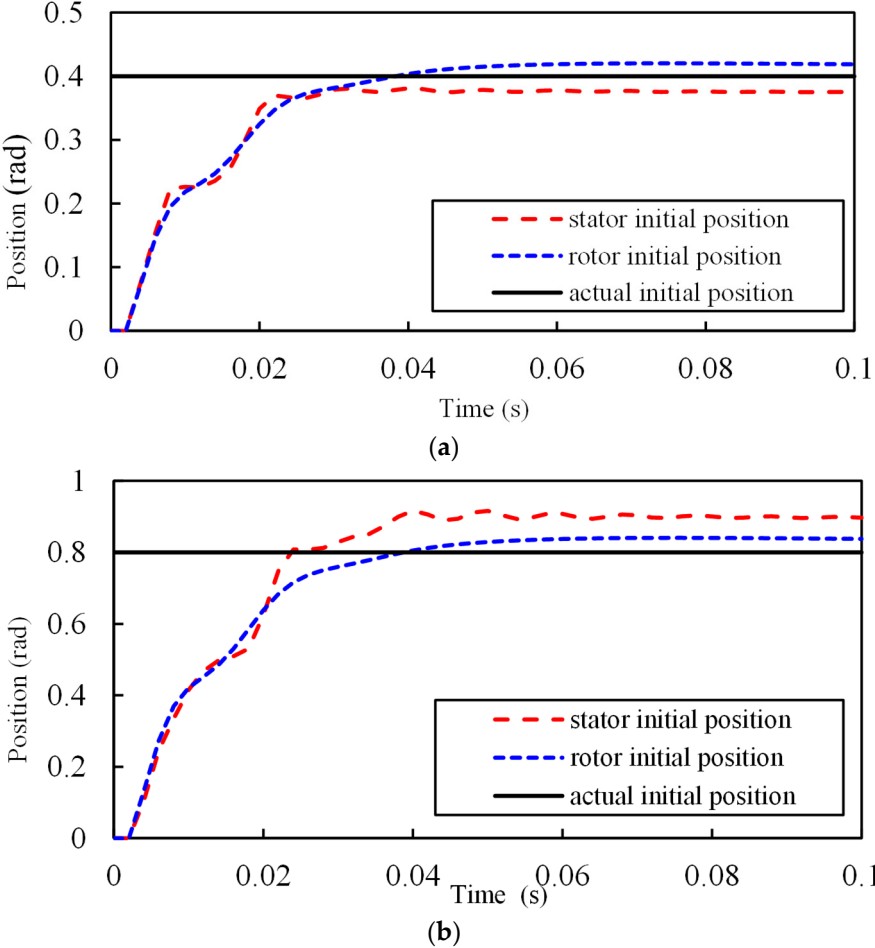

**Figure 10.** *Cont.*

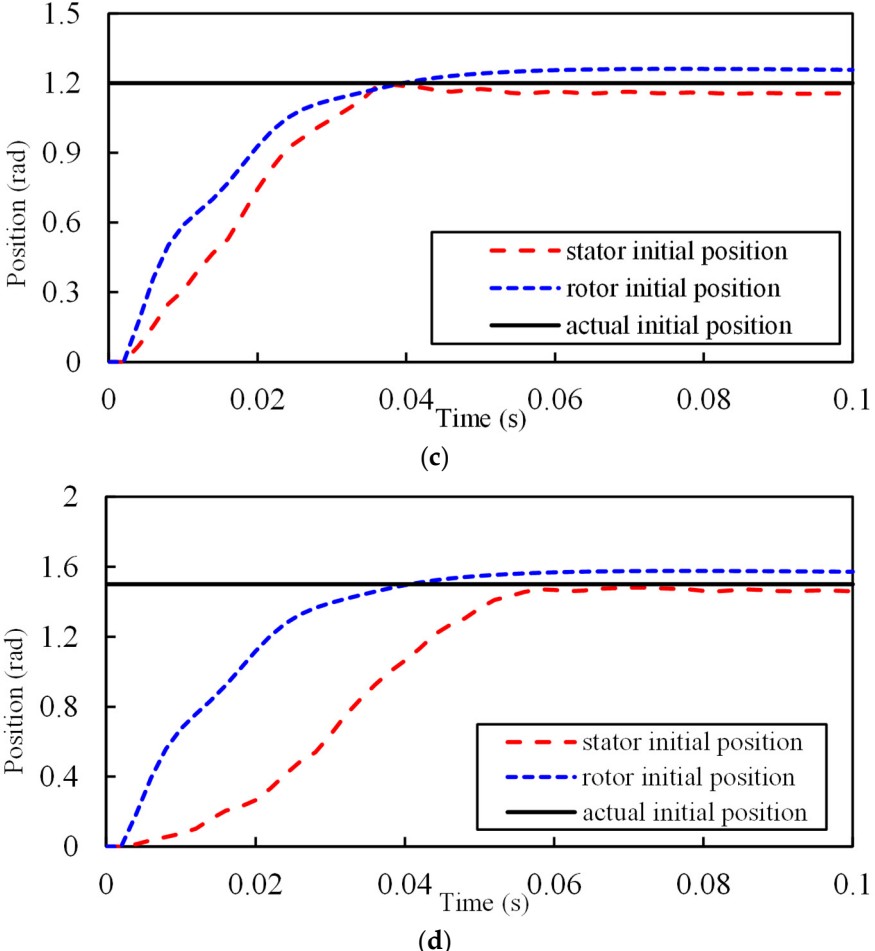

**Figure 10.** The initial position detection results. (**a**) The actual initial is 0.4 rad. (**b**) The actual initial is 0.8 rad. (**c**) The actual initial is 1.2 rad. (**d**) The actual initial is 1.5 rad.

The high-frequency pulsating voltage is injected into the stator windings, and both the initial positions of the stator winding and rotor windings are detected. As shown in Figure 10, under different actual initial position situations, the stator and rotor initial positions can both be detected. With the increase of the actual initial position, the detection results of the rotor windings are approximately the same, while that for the stator windings differs, which is mainly reflected in the response time increases slightly.

### 5.3. The Position Estimation Results

The position estimation results are given in this section. The startup experiments, speed step experiments, and load step experiments are carried out to validate the proposed sensorless vector control method. The estimation results from the same side are based on the differential inductance of the machine, and the results from the opposite side are based on the mutual inductance, as analyzed before.

#### 5.3.1. Startup Results

The startup experiment results with the reference speed set as 100 rpm are given in Figures 11 and 12, where the load torque is 3 Nm. The former estimation results are based on differential inductance, and the latter is based on mutual inductance, and their performances are compared in Table 2.

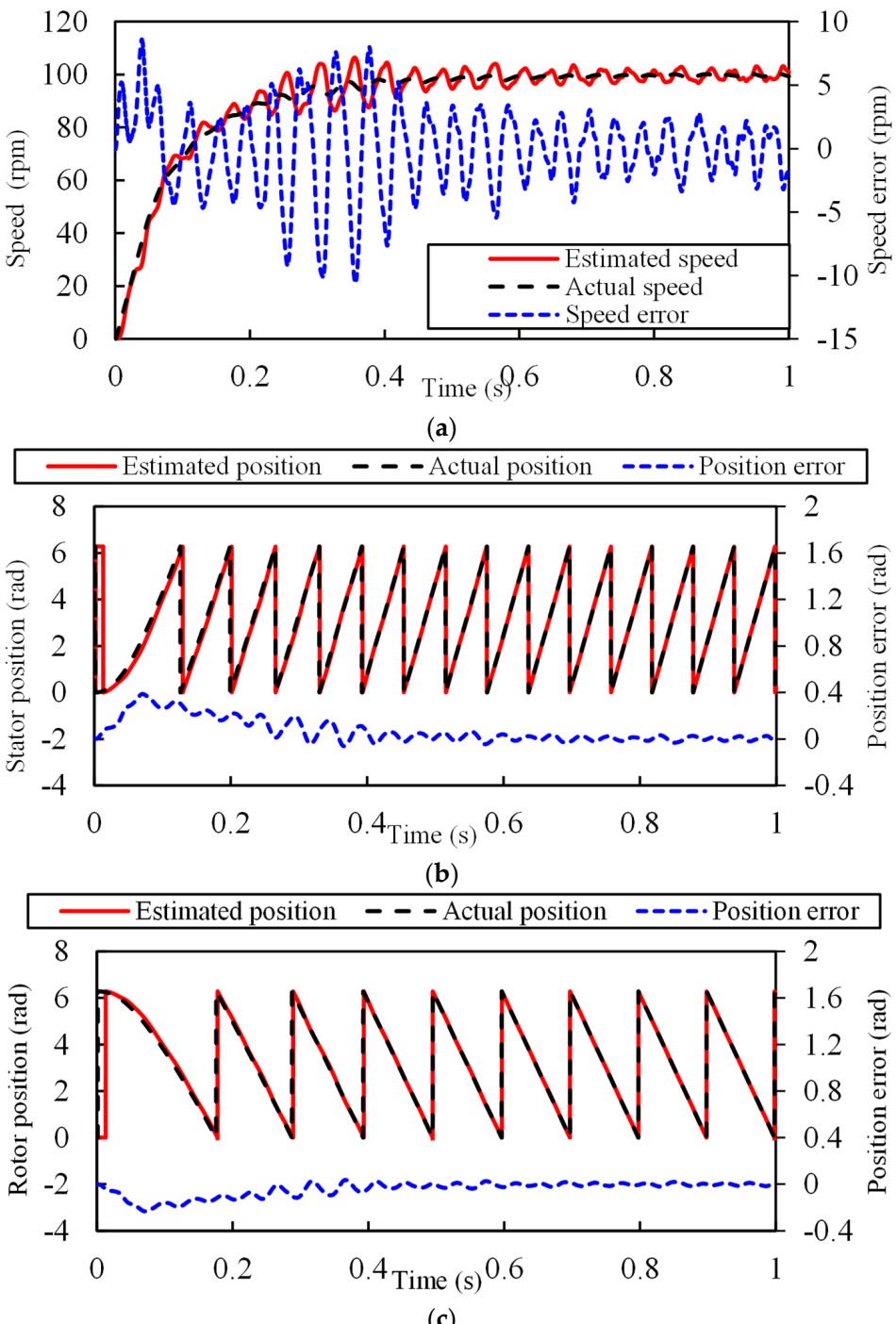

**Figure 11.** The startup results of estimating from the same side. (**a**) Speed. (**b**) Stator position. (**c**) Rotor position.

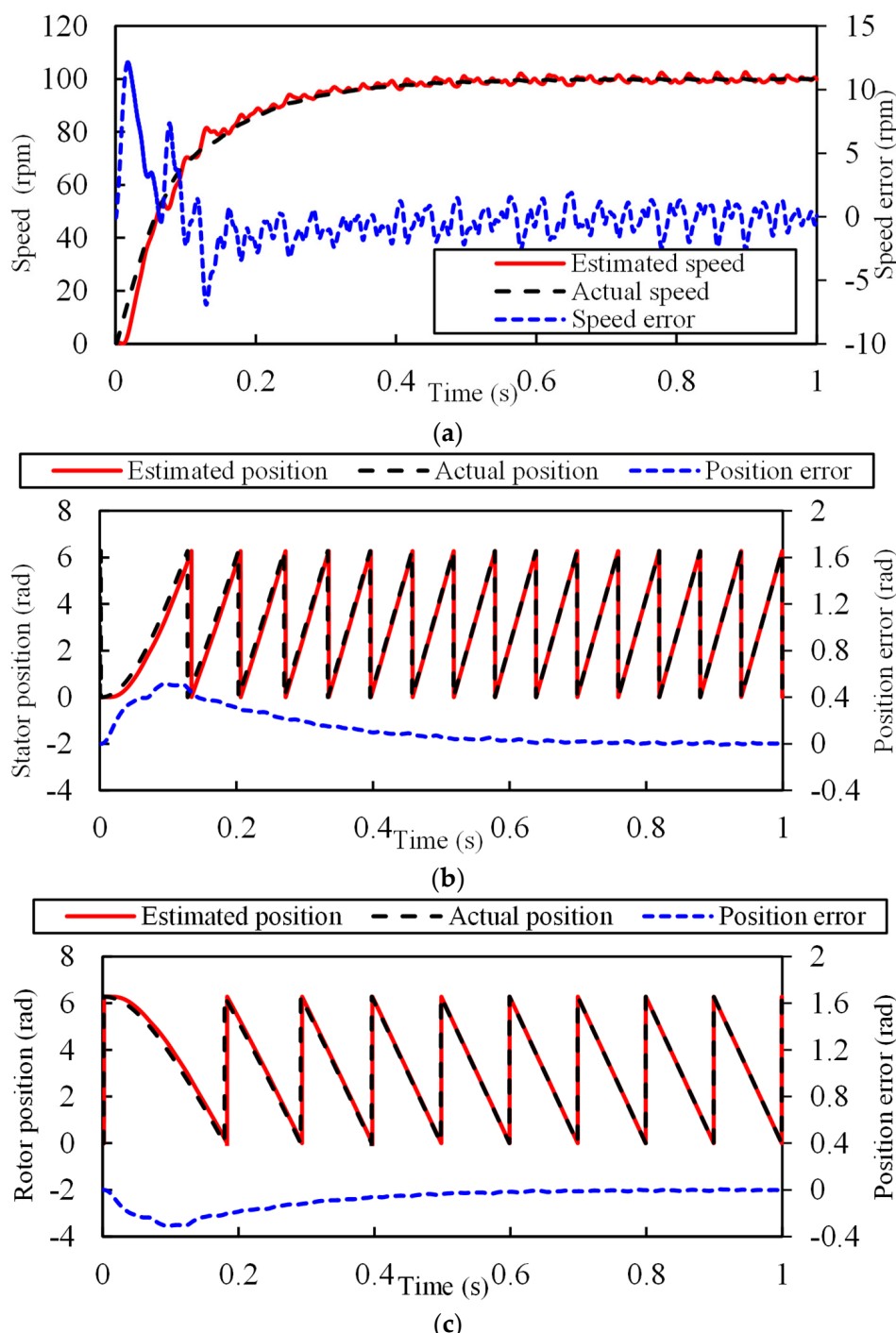

**Figure 12.** The startup results of estimating from the opposite side. (**a**) Speed. (**b**) Stator position. (**c**) Rotor position.

**Table 2.** Comparison of two estimating methods.

| | Estimating from the Same Side | | Estimating from the Opposite side | |
| --- | --- | --- | --- | --- |
| | Stator Windings | Rotor Windings | Stator Windings | Rotor Windings |
| Response time (s) | 0.5 | | 0.5 | |
| Steady speed ripple (rpm) | ±3.4 | | ±1.8 | |
| Steady position ripple (rad) | ±0.03 | ±0.02 | ±0.01 | ±0.008 |
| Maximum speed error (rpm) | 10.54 | | 12.16 | |
| Maximum position error (rad) | 0.39 | 0.23 | 0.52 | 0.31 |

As shown in the results in Figures 11 and 12, both methods can successfully estimate the positions of the machine. However, the performances of estimating from the opposite side are better than estimating from the same side, both in a steady performance and dynamic performance, which can be proved by the comparison based on several indicators given in Table 2. By estimating from the opposite side, the steady ripples of the speed and position are both reduced. Although the maximum speed and position errors increased, the speed fluctuation during the transition decreased.

### 5.3.2. Speed Step Results

The speed regulation performance of the algorithm is validated by the experiment results demonstrated in Figures 13 and 14. The speed reference value is changed from 100 rpm to 150 rpm in the experiment, and the load torque is kept at 3 Nm.

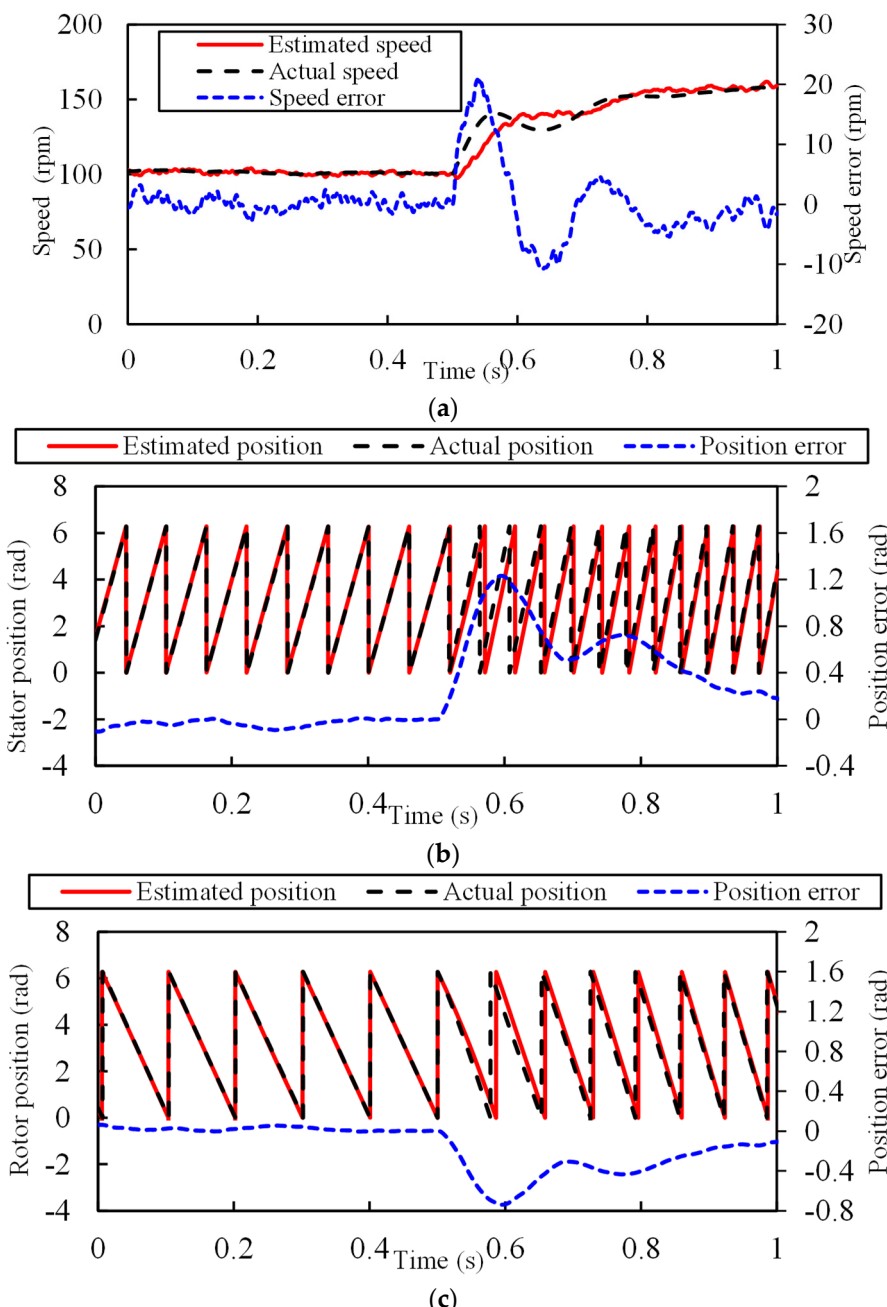

**Figure 13.** The speed step results of estimating from the same side. (**a**) Speed. (**b**) Stator position. (**c**) Rotor position.

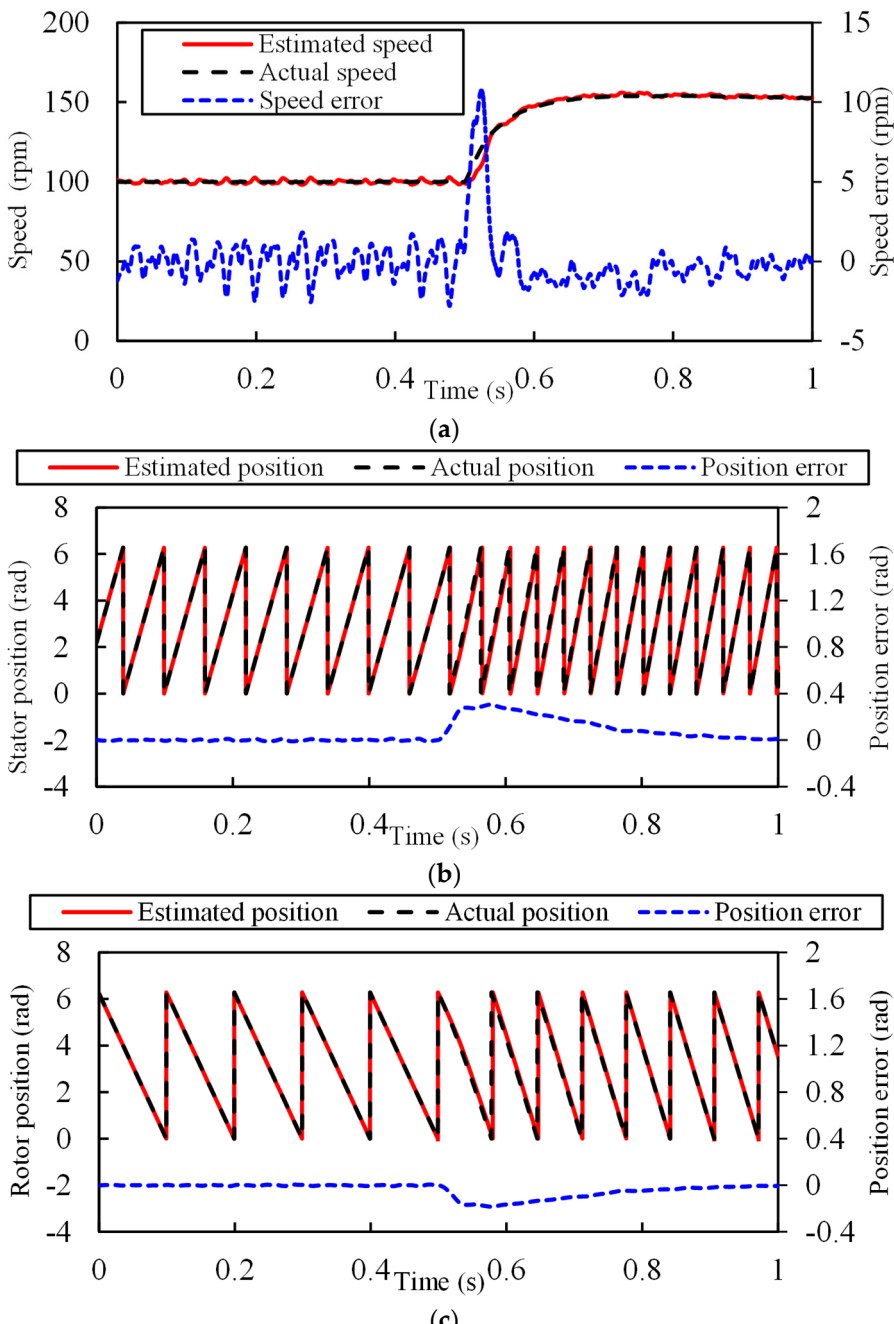

**Figure 14.** The speed step results of estimating from the opposite side. (**a**) Speed. (**b**) Stator position. (**c**) Rotor position.

The experiment results exhibited in Figures 13 and 14 are concluded and compared in Table 3, which demonstrates the superiority of the proposed position estimation method based on mutual inductance for the speed step condition. The response time of the system based on estimation from the opposite side is much shorter than estimation from the same side. Meanwhile, the steady speed and position ripples are also smaller, as well as the oscillation during dynamic response.

**Table 3.** Comparison of two estimating methods.

| | Estimating from the Same Side | | Estimating from the Opposite side | |
| --- | --- | --- | --- | --- |
| | Stator Windings | Rotor Windings | Stator Windings | Rotor Windings |
| Response time (s) | 0.5 | | 0.2 | |
| Steady speed ripple (rpm) | ±4 | | ±2 | |
| Steady position ripple (rad) | ±0.11 | ±0.3 | ±0.01 | ±0.006 |
| Maximum speed error (rpm) | 20.8 | | 10.75 | |
| Maximum position error (rad) | 1.23 | 0.74 | 0.31 | 0.18 |

### 5.3.3. Load Step Results

The loading performances are shown and compared in this section by the results in Figures 15 and 16. The load torque is changed from 3 Nm to 5 Nm at 0.2 s, and the speed reference is kept at 100 rpm.

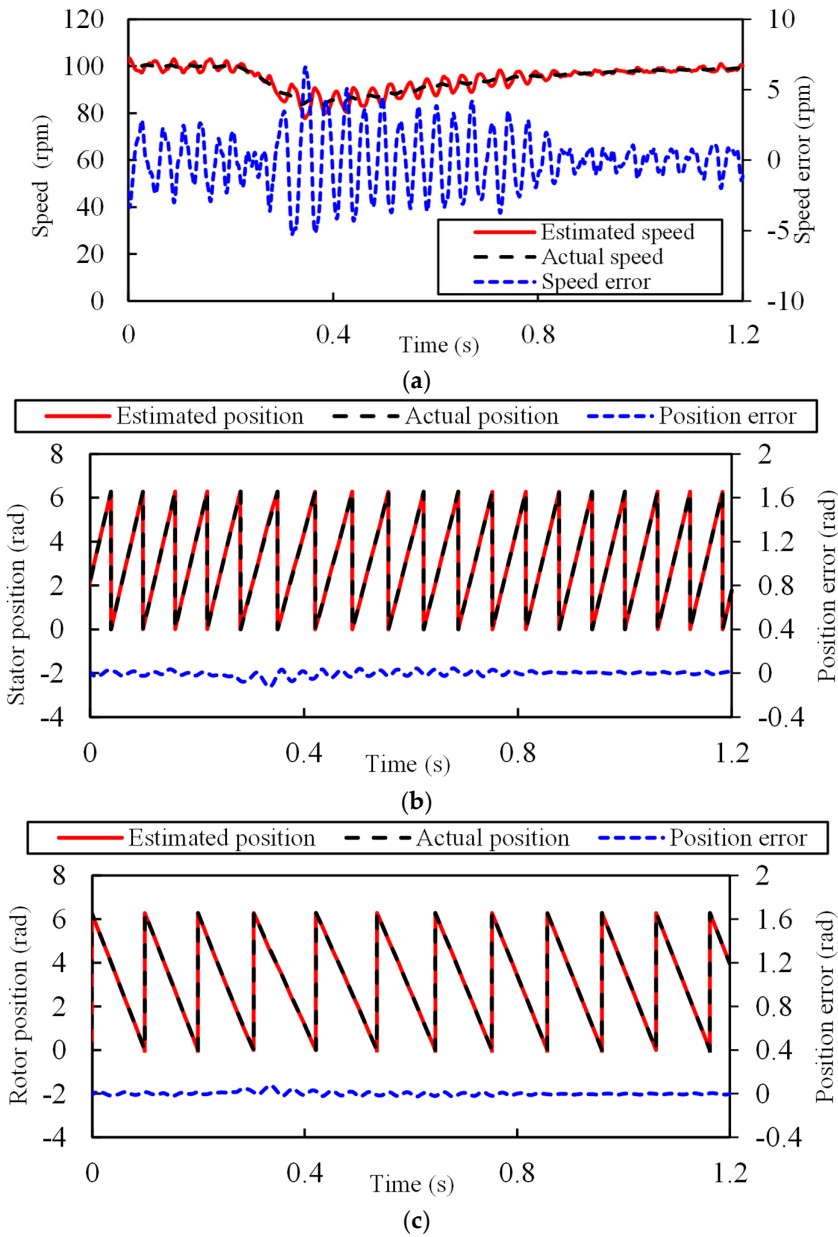

**Figure 15.** The load step results of estimating from the same side. (**a**) Speed. (**b**) Stator position. (**c**) Rotor position.

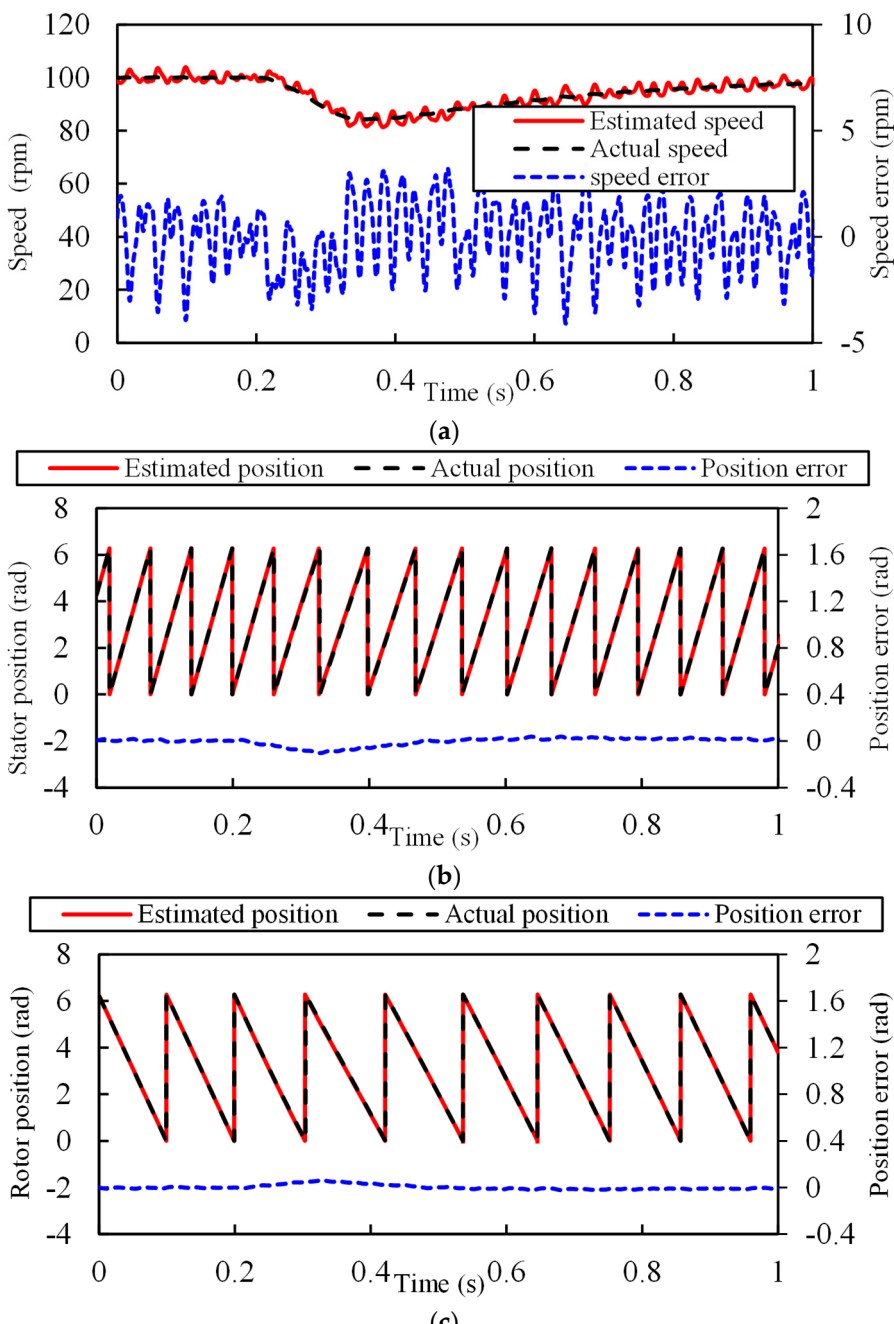

**Figure 16.** The load step results of estimating from the opposite side. (**a**) Speed. (**b**) Stator position. (**c**) Rotor position.

As shown in the figures and Table 4, when the load torque changes, the performance of estimating from the opposite side is also better than estimating from the same side. When estimated from the same side, the speed ripple during the transition is up to ±6.6 rpm, while that of estimating from the opposite side is ±4.1 rpm. Moreover, the transition time of the former is also longer by 0.25 s. The proposed method also behaves better in the steady speed and position ripple.

In conclusion, whether estimating from the same side or the opposite side, the sensorless vector control for the DA-FSPM based on HFI is effective. However, when estimating from a opposite side, i.e., extracting the position based on the mutual inductance, the performance is better, either on the steady or dynamic performance.

**Table 4.** Comparison of two estimating methods.

| | Estimating from the Same Side | | Estimating from the Opposite side | |
| | Stator Windings | Rotor Windings | Stator Windings | Rotor Windings |
| --- | --- | --- | --- | --- |
| Response time (s) | 0.95 | | 0.7 | |
| Steady speed ripple (rpm) | ±2 | | ±1.5 | |
| Steady position ripple (rad) | ±0.05 | ±0.03 | ±0.01 | ±0.01 |
| Maximum speed error (rpm) | 6.6 | | 4.1 | |
| Maximum position error (rad) | 0.12 | 0.07 | 0.1 | 0.06 |

## 6. Conclusions

In this paper, a novel HFI-based sensorless control method is proposed and implemented on the DA-FSPM prototype, which achieves better performance by extracting the position through the mutual inductance rather than the differential inductance of the machine. The high-frequency pulsating voltage is injected into one of the windings, and based on the high-frequency model derived in this paper, the initial positions detection method and position observer for both armature windings are further constructed. Both positions estimating approaches from the same side and opposite side are analyzed and implemented. As demonstrated in the experiment results, with the proposed mutual inductance-based position estimation method, the steady speed and position ripple is reduced. Moreover, the speed and load step experiments prove that the speed and position fluctuation and the response time during the transition are also reduced. In addition, the method has the potential to be extended to other machines with obvious mutual inductance, such as dual three-phase PMSMs.

**Author Contributions:** Formal analysis, L.W. and J.Y.; methodology, L.W. and J.Y.; validation, J.Y. and Z.Z.; investigation, Z.Z.; writing—original draft, J.Y.; writing—review & editing, L.W., Z.L. and S.H.; supervision, L.W. All authors have read and agreed to the published version of the manuscript.

**Funding:** This research was funded by "The National Natural Science Foundation of China, grant number 5197719 and 52225703".

**Data Availability Statement:** Not applicable.

**Conflicts of Interest:** The authors declare no conflict of interest.

## Nomenclature

| | |
| --- | --- |
| back-EMF | Back-electromagnet force. |
| DA-FSPM | Dual-armature flux-switching permanent magnet machine. |
| EKF | Extended Kalman Filter. |
| FSPM | Flux switching permanent magnet. |
| HFI | High-frequency injection. |
| PLL | Phase lock loop. |
| PMSM | Permanent magnet synchronize machine. |
| PI | Proportional-integral. |
| SMO | Sliding mode observer. |
| **u**, **i**, $\psi$ | Matrixes of voltages, currents, and flux linkages. |
| **R**, **L**, **M** | Matrixes of resistances, self-inductances, and mutual inductances. |
| **W** | Matrixes of cross-coupling coefficients. |
| $s$, $r$ | Parameters of the stator windings and rotor windings. |
| $s\_dq$, $r\_d1q1d3q3$ | Parameters of the *d-q* axes of the stator windings and *d*1-*q*1-*d*3-*q*3 axes of the rotor windings. |
| $srdq$ | Coupling component of the stator windings and rotor windings. |
| $p$ | Differential operator |
| $\omega_{se}$, $\omega_{re}$ | Stator and rotor electrical angular velocities. |
| $L_{sd}$, $L_{sq}$ | Self-inductances of the stator windings. |
| $L_{rd1}$, $L_{rq1}$, $L_{rd3}$, $L_{rq3}$ | Self-inductances of the rotor windings. |

| $M_{srd1}$ | Mutual inductances between the stator *d*-axis windings and the rotor *d*1-axis windings. |
|---|---|
| $M_{srq1}$ | Mutual inductances between the stator *q*-axis windings and the rotor *q*1-axis windings. |
| $M_{srd3}$ | Mutual inductances between the stator *d*-axis windings and the rotor *d3*-axis windings. |
| $M_{srq3}$ | Mutual inductances between the stator *q*-axis windings and the rotor *q3*-axis windings. |
| $M_{rd13}$ | Mutual inductances between the rotor *d*1-axis windings and *d3*-axis windings. |
| $M_{rq13}$ | Mutual inductances between the rotor *d*1-axis windings and *d3*-axis windings. |
| $f, f'$ | Variable in the real and estimated rotational coordinate system. |
| $\delta_{se}, \delta_{re}$ | Errors of the estimated angle and the real angle of the stator windings and rotor windings. |
| $\omega_h, \theta_h, V_c$ | Electrical angular speed, electrical angle, and amplitude of the injected voltage. |

## Appendix A

The coefficient matrix **A** in this paper is

$$\mathbf{A} = \begin{bmatrix} A_{11} & A_{12} & A_{13} & A_{14} & 0 & 0 \\ A_{21} & A_{22} & A_{23} & A_{24} & 0 & 0 \\ A_{31} & A_{32} & A_{33} & A_{34} & 0 & 0 \\ A_{41} & A_{42} & A_{43} & A_{44} & 0 & 0 \\ 0 & 0 & 0 & 0 & 0 & 0 \\ 0 & 0 & 0 & 0 & 0 & 0 \end{bmatrix} \tag{A1}$$

where

$$A_{11} = \hat{L_{sd}} \cos^2(\delta_{se}) + \hat{L_{sq}} \sin^2(\delta_{se}) \tag{A2}$$

$$A_{12} = \left( \hat{L_{sd}} - \hat{L_{sq}} \right) \cos(\delta_{se}) \sin(\delta_{se}) \tag{A3}$$

$$A_{13} = 5\hat{M_{srd1}} \cos(\delta_{re}) \cos(\delta_{se}) + 5\hat{M_{srq1}} \sin(\delta_{re}) \sin(\delta_{se}) \tag{A4}$$

$$A_{14} = 5\hat{M_{srd1}} \cos(\delta_{se}) \sin(\delta_{re}) - 5\hat{M_{srq1}} \cos(\delta_{re}) \sin(\delta_{se}) \tag{A5}$$

$$A_{21} = \left( \hat{L_{sd}} - \hat{L_{sq}} \right) \cos(\delta_{se}) \sin(\delta_{se}) \tag{A6}$$

$$A_{22} = \hat{L_{sq}} \cos^2(\delta_{se}) + \hat{L_{sd}} \sin^2(\delta_{se}) \tag{A7}$$

$$A_{23} = 5\hat{M_{srd1}} \cos(\delta_{re}) \sin(\delta_{se}) - 5\hat{M_{srq1}} \cos(\delta_{se}) \sin(\delta_{re}) \tag{A8}$$

$$A_{24} = 5\hat{M_{srq1}} \cos(\delta_{re}) \cos(\delta_{se}) + 5\hat{M_{srd1}} \sin(\delta_{re}) \sin(\delta_{se}) \tag{A9}$$

$$A_{31} = 3\hat{M_{srd1}} \cos(\delta_{re}) \cos(\delta_{se}) + 3\hat{M_{srq1}} \sin(\delta_{re}) \sin(\delta_{se}) \tag{A10}$$

$$A_{32} = 3\hat{M_{srd1}} \cos(\delta_{re}) \sin(\delta_{se}) - 3\hat{M_{srq1}} \cos(\delta_{se}) \sin(\delta_{re}) \tag{A11}$$

$$A_{33} = \hat{L_{rd1}} \cos^2(\delta_{re}) + \hat{L_{rq1}} \sin^2(\delta_{re}) \tag{A12}$$

$$A_{34} = \left( \hat{L_{rd1}} - \hat{L_{rq1}} \right) \cos(\delta_{re}) \sin(\delta_{re}) \tag{A13}$$

$$A_{41} = 3\hat{M_{srd1}} \cos(\delta_{se}) \sin(\delta_{re}) - 3\hat{M_{srq1}} \cos(\delta_{re}) \sin(\delta_{se}) \tag{A14}$$

$$A_{42} = 3\hat{M_{srq1}} \cos(\delta_{re}) \cos(\delta_{se}) + 3\hat{M_{srd1}} \sin(\delta_{re}) \sin(\delta_{se}) \tag{A15}$$

$$A_{43} = \left(\hat{L_{rd1}} - \hat{L_{rq1}}\right) \cos(\delta_{re}) \sin(\delta_{re}) \tag{A16}$$

$$A_{44} = \hat{L_{rq1}} \cos^2(\delta_{re}) + \hat{L_{rd1}} \sin^2(\delta_{re}) \tag{A17}$$

$$\hat{L_{sd}} = \frac{4\left(M_{rd13}^2 - L_{rd1}L_{rd3}\right)}{4L_{sd}M_{rd13}^2 - 30M_{rd13}M_{srd1}M_{srd3} + 15L_{rd3}M_{srd1}^2 + 15L_{rd1}M_{srd3}^2 - 4L_{rd1}L_{rd3}L_{sd}} \tag{A18}$$

$$\hat{L_{sq}} = \frac{4\left(M_{rq13}^2 - L_{rq1}L_{rq3}\right)}{4L_{sq}M_{rq13}^2 - 30M_{rq13}M_{srq1}M_{srq3} + 15L_{rq3}M_{srq1}^2 + 15L_{rq1}M_{srq3}^2 - 4L_{rq1}L_{rq3}L_{sq}} \tag{A19}$$

$$\hat{L_{rd1}} = \frac{15M_{srd3}^2 - 4L_{rd3}L_{sd}}{4L_{sd}M_{rd13}^2 - 30M_{rd13}M_{srd1}M_{srd3} + 15L_{rd3}M_{srd1}^2 + 15L_{rd1}M_{srd3}^2 - 4L_{rd1}L_{rd3}L_{sd}} \tag{A20}$$

$$\hat{L_{rq1}} = \frac{15M_{srq3}^2 - 4L_{rq3}L_{sq}}{4L_{sq}M_{rq13}^2 - 30M_{rq13}M_{srq1}M_{srq3} + 15L_{rq3}M_{srq1}^2 + 15L_{rq1}M_{srq3}^2 - 4L_{rq1}L_{rq3}L_{sq}} \tag{A21}$$

$$\hat{M_{srd1}} = \frac{2(L_{rd3}M_{srd1} - M_{rd13}M_{srd3})}{4L_{sd}M_{rd13}^2 - 30M_{rd13}M_{srd1}M_{srd3} + 15L_{rd3}M_{srd1}^2 + 15L_{rd1}M_{srd3}^2 - 4L_{rd1}L_{rd3}L_{sd}} \tag{A22}$$

$$\hat{M_{srq1}} = \frac{2\left(L_{rq3}M_{srq1} - M_{rq13}M_{srq3}\right)}{4L_{sq}M_{rq13}^2 - 30M_{rq13}M_{srq1}M_{srq3} + 15L_{rq3}M_{srq1}^2 + 15L_{rq1}M_{srq3}^2 - 4L_{rq1}L_{rq3}L_{sq}} \tag{A23}$$

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
