# Peer review of "High-Frequency Signal Injection-Based Sensorless Control for Dual-Armature Flux-Switching Permanent Magnet Machine"

_wevj, doi:10.3390/wevj14040085_

Round 1

Reviewer 1 Report

Some comments and suggestions are given here:

- In introduction, the main contribution and originality should be explained in more detail.

in the introduction,  i propose adding an organization section.

- The existing drawbacks and missing links need to be clearly highlighted in bullet points.

- Avoid abbreviations in titles and subtitles.

- Improve the quality of the Figure 7.

- Interpretation of implementation results need to be improved.

- Please compare the proposed method with recent results.

- I propose adding a list of the abbreviations.

- The conclusion isn’t accuracy.

Reviewer 2 Report

Reviewer statement for the manuscript:wevj-2281417

High-Frequency Signal Injection Based Sensorless Control for Dual-Armature Flux-Switching Permanent Magnet Machine

Summary of the Manuscript

The submitted manuscript addresses the position estimation method of DA-FSPM through an a sensorless control method based on HFI. If the proposed framework is successfully applied. It could have good application value. However, the manuscript also comes along with severe problems. The major deficiencies are the background, the presentation of the proposed methods and the report about the simulation experiment results. Taking into account these aspects I vote for a major and intensive revision of the current manuscript.

In introduction, it has to be reorganized. Whats the HFI method? What are its advantages? In addition, authors should states that the traditional HFI method has the problem of high frequency noise and filter abuse.The authors need to analyze the significant advantages of traditional algorithms such as sliding mode control and Kalman filtering.

To reduce noise, noise can be reduced directly by reducing the amplitude of high frequency signal, but at the same time, the signal to noise ratio will be reduced, which is not conducive to position identification.

The two most commonly used methods of HFI are rotating high frequency voltage injection and pulsating high frequency voltage injection. What's the difference between them?

The HFI usually requires a variety of filters to demodule the high-frequency current components containing rotor position information, and the use of a large number of filters reduces the dynamic performance of the system. How did the authors solve it?

Each control method has its own advantages and disadvantages, and has its own adaptation speed range, so hybrid methods are used to make up for the lack of single control method , such a composite control method is the current trend. What do the authors think of it?

The proposed method should be combined with practical application, simplifying the control structure and algorithm requirements as far as possible, so as to produce greater value in practical engineering. 

The conclusions need to be rewritten. Although authors state the extracted position information based on the mutual-inductance is more obvious than that extract from the saliency, the proposed method also has better steady and dynamic performance, which is proved by the analysis and experiments carried out on the prototype. Authors should summary major contributions, as well as specific measures.

Some formulas are garbled, and there are many basic equations that can be deleted.

Reviewer 3 Report

This paper deals with the sensorless control of a dual-armature flux-switching permanent magnet motor using high frequency signal injection technique. The introduction provides very limited bibliography references for further reading. A few important paper are missing.
Section 2 and 3 (Modelling of the DA-FSPM, HFI) are not very clear, and a few symbols are missing.
The paper has two big problems
1) not clear the new contribution,
2) the proposed design stategy should be applied to an automotive drivetrain or some like that, otherwise this paper is not in the field of interest of this journal (latest developments and knowledge about electric vehicles. Electric, Plug-in Hybrid, Hybrid Electric, or Fuel Cell Vehicle). In my opinion the paper is this form is out of scope, and is not suitable to be published in this journal.

Minor:
* fig. 2: a few symbols are missing
* fig. 3: a few symbols are missing
* eq (32), (33) a few symbols are missing

Round 2

Reviewer 1 Report

The authors have improved the article according to the recommendations. The article can be accepted for publication.

Reviewer 2 Report

Authors have addressed my concerns. It can be accepted now.

Reviewer 3 Report

Good work. In my opinion the paper is now readry to be published.